# Media representations of China's inclusive education: A corpus-assisted critical discourse analysis

Yating Yu[1], Gaoqiang Lu[2], Kuen Fung Sin[3]*, Zhiying Yu[4]

**1** Department of Communication, The University of Macau, Macau, China, **2** Department of Linguistics and Translation, City University of Hong Kong, Kowloon, Hong Kong, **3** Institute of Special Needs and Inclusive Education, The Education University of Hong Kong, Tai Po, Hong Kong, **4** Faculty of Humanities and Social Sciences, Beijing Normal University & Hong Kong Baptist University United International College, Zhuhai, Guangdong Province, China

\* kfsin@eduhk.hk

**Data Availability Statement:** The minimal underlying data are publicly available from the OSF repository (https://osf.io/sk5ne/).

**Funding:** The authors received no specific funding for this work.

## Abstract

The Chinese government aspires for inclusive education to serve as an exemplary model in addressing educational inequity issues and establishing a responsible global image as a major power. Nevertheless, there has been limited focus on China's news media coverage concerning its inclusive education initiatives aimed at international audiences. To bridge this gap, this study employed a corpus-assisted critical discourse analytic approach to examine China's self-image in relation to its inclusive education endeavours. Seventy-three English-language news articles from the official channel were retrieved for meticulous line-by-line concordance analysis. The results indicate that out of the 520 co-occurring instances, they can be broadly categorised into four groups: efforts to develop inclusive education (65.4%); consensus on benefits of inclusive education (24.5%); challenges faced by inclusive education (4.8%); and others (5.4%). This study illuminates the effective utilisation of official media in the image construction of inclusive education in China.

## 1. Introduction

Inclusive education (IE) is an education model that focuses on equity, access, opportunity, and rights [1]. It has brought significant changes to mainstream schools [2], one of which is the increase in teacher diversity, as students with special educational needs are educated in regular classrooms [3]. In mainland China, IE mainly targets students with seven types of disabilities: intellectual disability, hearing impairment, visual impairment, mental disorder, speech and language impairment, physical disability, and multiple disabilities [4]. Mainstream schools that accept these students are usually called "IE schools". However, owing to the comparatively later introduction of IE in China in contrast to developed nations and the nuanced nature of regional culture, which presents difficulties in ensuring simultaneous support to all individuals, there remains a notable level of uncertainty regarding the establishment of an inclusive school environment and the effective implementation of IE [5].

**Competing interests:** The authors have declared that no competing interests exist.

The core concept of IE is to allow students with disabilities or special needs to share a classroom with other students, learning and communicating together. Such an educational environment can foster communication and collaboration among students, nurturing friendship, understanding, and respect. Hence, this aligns with IE, aiding China in accomplishing notable advancements in implementing IE. Simultaneously, the implementation of IE also offers increased support and care for students with disabilities or special needs, presenting a harmonious portrayal of society, which also contributes to the national image.

The concept of a national image can be divided into two types: self-image and other-image [6, 7]. Other-image typically involves paying attention to news reports from other countries. Self-image, on the other hand, often relies on official media to shape a positive image. Official media, particularly state-controlled media, is seen as an effective tool for disseminating ideology and shaping overseas perceptions [8, 9]. For instance, China utilises English media to communicate its contributions to disaster relief, medical aid, and peacekeeping during the COVID-19 pandemic [10]. By taking this approach, it presents itself as a responsible power and a valuable partner to the international community, demonstrating its willingness to make positive contributions to global stability and development [11, 12].

Returning to IE, the Chinese government introduces its educational policy and teaching philosophy to the world through foreign-language media to enhance other countries' understanding of China's national conditions. This also helps to establish friendly relations between people and strengthen international cooperation and exchange. In this way, China actively conveys its vision, concept, and commitment to the world, which helps to promote the understanding and recognition of China by the international community and establish China's image as a responsible power. However, no research can be found to focus on the reporting of IE by China's official foreign-language news media, which is not conducive to revealing the international image that China tries to show to the international community. Based on this consideration, this paper designs two research questions:

1. What are the underlying discourses in the media representations of IE in China's English-language news media?

2. What discursive strategies are employed to construct these discourses?

We employed corpus-assisted critical discourse analysis to address these two questions. Firstly, this linguistic approach enabled us to examine a substantial amount of data, thereby mitigating researcher bias. Secondly, it allowed us to observe how the state-controlled media utilised linguistic choices and discursive strategies to "produce a particular version of the world" and shape its national image [13]. Discursive strategies refer to "a more or less intentional plan of practices (including discursive practices), adopted in order to achieve a particular social, political, psychological, or linguistic goal. Discursive strategies are located at different levels of linguistic organization and complexity" [14]. Investigating discursive strategies is crucial in this study as they aid in comprehending the communication patterns employed by the state-run media to construct persuasive discourses and uncover power dynamics. In the subsequent sections, we will present our literature review and methodology, followed by our findings, discussion, and conclusion.

## 2. Literature review

### 2.1. National image and state-run news media

Research indicates that the Chinese government utilises its state-run news media to enhance its national image and communicate its messages globally. For instance, Yu, Tay, and Yue [12]

employed corpus-assisted critical discourse analysis to explore the representation of China in its state-run news media during the COVID-19 outbreak. They discovered that China was depicted as a collaborator, a fighter, and a victim. Similarly, Yu [15] conducted critical discourse analysis to investigate how the Chinese government employed its state-controlled news media to convey geopolitical messages and counter unwarranted foreign hostility towards China when COVID-19 was initially discovered in the country. She uncovered that the media constructed a "discourse of resistance" in their crisis communication. Chan and Yu [10], in their investigation utilising positive discourse analysis, found that China employed its state-run news media to foster solidarity globally during COVID-19 and enhance China's international relations. Their findings revealed China's dominant diplomatic concept of building a global community of health for all.

In recent years, IE in China has witnessed rapid development. In their review article, S. Q. Xu et al. [16] primarily discussed the initiative known as the "Learning Resource Class" in China. This initiative has been in place since the 1980s and has gradually expanded with the support of the government. Its primary objective is to enhance the enrolment rate of students with disabilities in mainstream schools and provide them with educational opportunities. Deng and Poon-McBrayer [17] provided an overview of the history, current status, and future prospects of IE in China. They highlighted that while China has made significant progress in special education over the past decade, challenges in implementing IE persist, such as a shortage of trained professionals, inadequate resources, and an imperfect education system. To tackle these issues, the Chinese government has implemented various measures, including bolstering teacher training, increasing investments in education, and enhancing laws and regulations [17].

Several empirical studies have also examined IE in China, including the works of Florian and Linklater [18], Xie and Zhang [2], and Cui [19]. Florian and Linklater [18] conducted research using a mixed-method approach, investigating data on a new teacher training course designed to provide IE training to mainstream classroom teachers. They presented the results of a new initial teacher education course that is based on the notion that, when students encounter challenges, it is more effective to leverage teachers' knowledge and skills rather than solely considering their qualifications to teach in inclusive classrooms. Xie and Zhang [2] employed a questionnaire survey method involving 300 teachers from primary schools in Beijing. The research revealed that teachers involved in IE demonstrated greater support and commitment to its implementation in schools. Cui [19], on the other hand, adopted a case study approach to elucidate the challenges faced by IE in China, particularly the multiple barriers encountered by children with disabilities in advocating for IE. To address the structural discrimination that leads to segregation and exclusion in the classroom, she identified existing structural inequalities, fosters momentum for future changes, and provides a compelling example of parent advocacy.

In summary, existing research on IE in China encompasses macro-level perspectives, including the international context and policies, as well as micro-level perspectives, focusing on the current situation and challenges. However, no studies have been found that employ corpus and discourse analysis of China's English-language news media reports to examine how IE in China is represented on the international stage. This research could offer a distinctive viewpoint, unveiling the influence of news media in shaping foreign public perceptions of IE in China. Such insights are crucial in understanding an essential facet of China's global image.

## 2.2. Corpus-assisted critical discourse analysis

The corpus-assisted critical discourse analytical approach is a valuable method for examining news media, which this study employs. The synergy between corpus linguistics (CL) and

critical discourse analysis (CDA) represents a relatively recent approach to address the relevant criticisms of these two respective methods [20, 21]. CL has a descriptive and quantitative nature, enabling the examination of vast amounts of data [22, 23]. However, CL's emphasis on quantification has resulted in accusations of "bean counting" [24]. There is a need to go beyond the surface level of analysis. On the other hand, CDA possesses a critical and qualitative nature. Some scholars label its analysis as "cherry-picking" to align with the researcher's agenda and argument [23]. The combination of both approaches can complement each other and compensate for their shortcomings. CL can assist in investigating extensive data and revealing the language patterns that describe a social actor or phenomenon. CDA can provide in-depth analysis that surpasses the superficial level of language patterns. By combining both approaches, corpus-assisted CDA has proven to be a valuable method for in-depth investigation of extensive data over the past decade [25].

Some scholars have employed the corpus-assisted CDA approach to investigate their media data. For instance, utilising this approach, Baker [26] analysed the media portrayals of Islam and Muslims in the news media and discovered that the British press tended to associate them with extreme beliefs. Similarly, Yu [20] employed this approach to examine media depictions of single older women in China and discovered that the term "leftover women" was linked to a social phenomenon, neologism, and various activities in the news. Likewise, Zhang and Yu [27] utilised this method to compare the representations of the concept of the Chinese Dream in three different press outlets: Mainland China, Hong Kong, and the United States. They discovered that China's state-controlled media actively utilised this concept to promote Chinese national images, the American news media conveyed a negative perception of this concept, and the Hong Kong-based news media remained neutral.

The previously mentioned studies have provided valuable insights into how the corpus-assisted CDA approach can be employed to examine media data. However, there is an opportunity for our study to contribute further. Baker [26] has highlighted the importance of corpus techniques in enhancing the objectivity of CDA, as well as the role of CDA in enriching the descriptive nature of corpus-driven analysis. Nevertheless, he primarily argued for the need to interpret language patterns critically in order to identify language bias. Yu [20] utilised van Leeuwen's sociosemantic approach from CDS and Sinclair's concept of meaning shift units from CL, but she disregarded the discursive strategies employed by the media to construct representations of marginalised groups. While Zhang and Yu [27] investigated the discursive strategies used to shape the Chinese dream, their focus was limited to specific linguistic indicators using the corpus approach, such as the reporting verb "said". In contrast to previous studies, our research presents a model for conducting corpus-assisted CDA to analyse how the media portrays education, particularly IE, in order to shape a nation's image. The following section will demonstrate the specific application of the corpus-assisted critical discourse method, specifically focusing on the semantic prosody, as employed in our study.

## 3. Method

To establish a corpus examining the portrayal of IE in China's English-language news, researchers utilised Factiva, an online database of news articles, to retrieve relevant news articles from six newspapers considered to be official Chinese media at both the national and local levels (Table 1). Factiva is a commercial news database owned by Dow Jones, which has been widely utilised by researchers across diverse disciplines [28, 29]. Factiva is available at https://professional.dowjones.com/factiva/. Users need to subscribe to Factiva with an amount of subscription fee in order to use it. The minimal underlying data are publicly available from the OSF repository https://osf.io/sk5ne/. The selected newspapers represent the prominent state-

**Table 1. Details about the news article selection.**

| Newspapers | N (%) | Tokens (%) |
|---|---|---|
| 1. China Daily | 39(53.42%) | 85,935(82.57%) |
| 2. Xinhua News Agency | 27(36.99%) | 13,411(12.89%) |
| 3. Shanghai Daily | 4(5.48%) | 2845(2.73%) |
| 4. Global Times | 2(2.74%) | 1565(1.50%) |
| 5. People's Daily | 1(1.37%) | 324(0.31%) |
| Total | 73(100%) | 104,080(100%) |

controlled news media [30] in terms of user coverage [31, 32]. For instance, *China Daily* is the largest English news portal catering to foreign audiences in China [33], offering news, business information, and other content [34]. *Xinhua News Agency* holds significant influence as one of China's most prominent media outlets [35]. During the data collection stage, *Shenzhen Daily* was also included, but no news articles were retrieved. Therefore, it was omitted from the table. These newspapers were chosen due to their ability to provide a broad reflection of the Chinese government's position on the subject matter.

The search format employed for data retrieval was "inclusive education*" (utilising the wildcard asterisk * to encompass both the singular and plural forms). Only news articles that featured these keyword phrases in the title and/or the opening section were gathered, resulting in a total of 137 pertinent articles. Furthermore, two authors independently conducted a meticulous assessment to evaluate the pertinence of all the news articles. They also carried out an inter-rater reliability analysis. The outcome of the Kappa Coefficient test, with a value of 0.98, suggests a substantial level of concurrence between the authors [36]. After conducting manual screening to extract content pertaining to IE in China, 73 articles remained, constituting a total of 104,080 tokens. The publication dates of these news articles range from May 12, 2006, to June 10, 2023, with the former denoting the date of the initial relevant article, and the latter indicating the commencement date of data collection by the authors.

After completing the data cleaning stage, the data was imported into corpus analysis tools, specifically WordSmith v.6.0 [37] for generating collocates (i.e., recurring words near the search term "inclusive education*"), and ConcGram v. 1.0 [38] for generating concordance lines. Firstly, by employing a span of five words ("L5 and R5") based on Sinclair et al. [39] and utilising the Concord function in WordSmith v.6.0, lemmatised collocates were retrieved for "inclusive education*". Please note that only content words that occurred more than once are deemed as "valid collocations" [40], with function words being manually excluded [20]. Secondly, ConcGram v. 1.0 assists researchers in generating concordance lines that display the co-occurrence of the term "inclusive education*" with frequent collocates. Subsequently, the findings will be exported to an Excel file for meticulous line-by-line semantic prosody analysis. Semantic prosody (also known as discourse prosody) is characterised as "a feature which extends over more than one unit in a linear string" [41] and "a consistent aura of meaning with which a word is imbued by its collocates" [42]. For example:

Hu Meifeng, principal of the special education school of Beichen District, said that more than 80 disabled children from the district have joined the education program in the compulsory education phase, and more than 20 disabled people have found *jobs* after the **inclusive education** [43].

This example represents one outcome of the co-occurrence between "inclusive education" and "jobs". The phrase "20 disabled people have found jobs" conveys a positive sentiment associated with job opportunities. Consequently, it was categorised as "job opportunities after

inclusive education". Employing a similar approach, researchers analysed and classified all 520 instances of co-occurrence, ultimately organising them into distinct discourses and semantic prosodies, which helped to answer the first research question. To address the second research question concerning discursive strategies in the construction of these discourses, we employed Wodak's [14] discourse-historical approach (DHA) within the framework of CDA. This approach encompasses five strategies: nomination, predication, argumentation, perspectivisation, and intensification/mitigation, which are employed by speakers/writers to shape a discourse. Nomination involves constructing the role of a social actor. Predication describes the actions or attributes of a social actor. Argumentation utilises various types of topoi (singular topos) to justify or challenge a claim. Perspectivisation offers the speaker/writer's viewpoint to demonstrate their level of involvement or detachment. Intensification/mitigation is employed to adjust the illocutionary force of a sentence. By employing Wodak's DHA, we can introduce a critical element to examine how the media constructed its discourses.

## 4. Results

From the 520 co-occurring results, they can be broadly categorised into four groups, which indicate the three most frequently occurring discourses (Table 2). The subsequent sections will present a detailed analysis of the three primary discourses: "efforts to develop inclusive education", "benefits of inclusive education", and "challenges faced by inclusive education". The search term "inclusive education*" is highlighted, and its frequent collocates are displayed in italics in the examples. The discursive strategies employed to construct these discourses include nomination, predication, argumentation, and perspectivisation [14]. The "others" category encompasses various semantic prosodies that occur infrequently and therefore do not contribute significantly to any one discourse. For instance, we classify the first sentence below as expressing the semantic prosody of "high hopes of parents of disabled students for IE", and the second sentence as conveying the "importance of IE". However, both semantic prosodies

**Table 2. Semantic prosodies on the term "inclusive education*".**

| Discourse | Semantic Prosody | N(%) |
|---|---|---|
| 1. Efforts to develop inclusive education (65.4%) | a. Government's efforts to develop inclusive education | 214 (41.2%) |
| | b. School's efforts to develop inclusive education | 61(11.7%) |
| | c. Pilot inclusive education projects | 40(7.7%) |
| | d. People's efforts to develop inclusive education | 16(3.1%) |
| | e. Greater investment in inclusive education | 9(1.7%) |
| 2. Benefits of inclusive education (24.5%) | a.Consensus on the benefits of inclusive education | 75(14.4%) |
| | b. Job opportunities after inclusive education | 25(4.8%) |
| | c. Children can learn from each other | 11(2.1%) |
| | d. A better learning environment | 6(1.2%) |
| | e. Impact of inclusive education on confidence and independence | 5(1.0%) |
| | f. Integrate into society | 5(1.0%) |
| 3. Challenges faced by inclusive education (4.8%) | a. Challenges faced by inclusive education | 14(2.7%) |
| | b. Limited availability | 7(1.3%) |
| | c. Lack of capable teachers in inclusive education system | 4(0.8%) |
| 4. Others (5.4%) | a. Others | 28(5.4%) |
| **Total** | | **520 (100%)** |

occur only once. In the following subsections, we will analyse each of the three most frequently occurring discourses in detail.

1. Parents of disabled students *have* high hopes for **inclusive education**, hoping that this could help their children [44].

2. China has highlighted the *importance* of **inclusive education** in its rules on education for the disabled, there is a long way to go for the policy to be translated into tangible benefits [45].

## 4.1. Efforts to develop inclusive education

Five semantic prosodies contribute to the first discourse by occupying a significant proportion of 65.4% (Table 2). This sizeable percentage illustrates the favourable impression that official media outlets endeavour to create concerning the initiatives taken for IE, encompassing those by the government, schools, and individuals. These endeavours entail pilot projects in IE and increased investment.

> **Example 1.** The *government*'s plan for **inclusive education** aims to have 95 percent of children with disabilities enrolled in mainstream schools by 2020 [46].

First and foremost, the support from the government has garnered the most attention (41.2%). When the government sets proactive goals, such as integrating "95 percent of children with disabilities" into mainstream schools through IE programmes, it demonstrates its commitment to promoting social inclusion and equality, as Example 1 shows. The number "95 percent" is an argumentation strategy (specifically a topos of numbers) that the media used to support its claims and generalisation with statistical evidence [14]. This programme aims to integrate disabled children into the mainstream educational system, eliminating their isolation and discrimination [19]. The government has established clear objectives and set a specific timeframe (i.e. by 2020), indicating their active and conscious efforts to promote IE in a planned manner [19]. This initiative not only provides better educational opportunities for disabled children but also contributes to the creation of a more inclusive social environment.

It is worth noting that many schools, including kindergartens, primary schools, middle schools, and high schools, are public schools managed by the government [47]. Therefore, it is only through the implementation of pertinent government policies (e.g. the "Learning in Regular Classrooms" initiative) that these schools will adopt proactive measures to accommodate children with special needs [17]. Without government support and guidance, schools are likely to be hesitant about accepting special needs children (which will be discussed below). This implies that while the government formulates comprehensive IE plans, it also needs to ensure effective implementation at the school level. This requires collaboration between the government and schools to provide the necessary resources, training, and support, enabling schools to meet the needs of special needs children and provide them with an inclusive educational environment. Through government policy guidance and comprehensive efforts, public schools can become significant drivers in achieving IE goals and providing equal educational opportunities for special needs children. The media coverage of these efforts helps shape a positive and responsible image of the government internationally, as Example 1 shows.

> **Example 2.** In China, more and more schools and welfare *institutions* are promoting **inclusive education** for children of different ages and with different health conditions, and help those with different functional needs to enjoy equal access to education and integrate into society in a better way [48].

With the backing of government policies, the school system has begun undergoing restructuring and offering appropriate assistance to students with diverse learning needs in mainstream schools. This necessitates practitioners and researchers to reconsider their thoughts and practices, redeveloping the curriculum and teaching methods to ensure equal educational opportunities for all [49]. It also signifies a shift in attitudes towards students with special educational needs, from schools to teachers, treating them with acceptance [49]. Furthermore, IE students typically have a range of physical or psychological disabilities in comparison to regular students, thus necessitating schools to exert additional efforts in special education. Example 2 indicates that an increasing number of schools are offering support and assistance to "children of different ages and with different health conditions" and helping those "with different functional needs" to access education on a fair and inclusive basis, enabling them to integrate more effectively into society.

Schools encourage and facilitate social interaction and friendship among students with special needs and other students, thereby dismantling isolation and discrimination and fostering a more inclusive and amicable school environment. Schools have a vital role to play in promoting IE. Lastly, schools also strive to nurture students' appreciation and reverence for diversity and inclusivity, by promoting equality and inclusive values through educational initiatives and awareness-building endeavours. In conclusion, schools assume a pivotal role in advancing the objectives of IE through the provision of inclusive environments, the acceptance of students with special needs, the provision of support and assistance, the promotion of social interaction, understanding, and respect. They can make noteworthy contributions towards achieving the goals of IE. The use of the adjective phrase "more and more" to describe "schools and welfare institutions" acts as an argumentation strategy (specifically a topos of numbers) [14] to emphasise the growing number of organisations and people involved in enhancing China's educational system in Example 2.

**Example 3.** This is because at schools, teachers such as George like to think that they are *working* toward an **inclusive education** for all; whatever a pupil's particular needs, difficulties, talent and/or disability [50].

In China, IE is primarily implemented by educators and IE teachers who serve as pioneers in promoting IE practices. They augment their understanding and proficiency in IE by taking part in professional training courses, with the aim of fostering the complete development and involvement of students in the teaching-learning process [51]. They devise and execute adaptable instructional strategies for children with disabilities, guaranteeing individualised educational plans that offer equitable opportunities for every student [52]. Additionally, educators exert extra efforts, such as advocating for the acceptance of special education students regarding their "needs, difficulties, talent and/or disability" by both teachers and peers, to achieve educational equity, as Example 3 indicates. The verbal phrase "working toward an inclusive education for all" serves as a predication strategy [14] to emphasise the work that educators have undertaken for students with special needs, highlighting the efforts made by the Chinese professional community.

**Example 4.** From 2009 to 2015, "Save the Children" managed to *pilot* **inclusive education** projects at primary schools in Sichuan and Yunnan provinces, and Xinjiang Uygur Autonomous Region, transforming old special schools into resource centres, and training teachers and parents [53].

**Example 5.** Liu contacted "Save the Children", a global organisation for child development and protection, which has successfully *piloted* **inclusive education** programs in China for years, to use his expertise to help disabled children [53].

**Example 6.** Far greater investment *needs* to be made in **inclusive education** to enable all children–regardless of their ability–to attend and learn at mainstream schools [45].

Furthermore, because of the relatively limited educational resources in remote areas in some regions, such as "Sichuan", "Yunnan", and the "Xinjiang Uygur Autonomous Region", and the insufficient number of special education institutions and professional teachers, many children with disabilities are unable to access appropriate educational services (Example 4). Consequently, the inclusion of children with disabilities in regular schools has emerged as an effective solution, in line with the educational equity policy advocated by the central government [54]. The success and sustainable development of these projects necessitate greater investment to ensure that all students can access high-quality education. Liu, mentioned in Example 5, is one of the many individuals in China who are dedicated to helping children with disabilities. The media used the phrase "use his expertise to help disabled children" to describe him and implied the existence of a sizeable volunteer community for children with special needs in China, illustrating the function of predication [14]. In Example 6, the term "greater investment" signifies the necessity for stronger support from various sectors of society for IE. This term serves as a perspectivisation strategy [14], contributing to the cultivation of a positive and responsible image for those involved and expressing the commitment and involvement of Chinese society in promoting IE [55].

To conclude, the first discourse presents a comprehensive overview of the initiatives undertaken by various sectors of Chinese society to promote IE projects. It emphasises the backing received from the government at the policy level, as well as the role of schools in fostering and facilitating social interaction and friendships among students with special needs and their peers. The ultimate goal is to dismantle feelings of isolation and discrimination and cultivate a more inclusive and amicable school environment. Educators and teachers are recognised as the direct executors of these projects, showcasing their care and empathy, and striving to enhance equal opportunities for interaction between students and individuals with disabilities. These endeavours have led to the establishment of an integrated three-tier system encompassing the "government–schools–teachers".

## 4.2. Benefits of inclusive education

The government, schools, and teachers' endeavours in promoting projects for IE arise from a shared agreement regarding its apparent advantages. Our investigation reveals that the advantages of IE are a topic of frequent media reports manifested in six semantic prosodies (24.5%), contributing to the second discourse (Table 2). IE encompasses numerous benefits as it nurtures social development and prosperity through the provision of equal educational opportunities [56]. This educational approach can grant equal rights to students with special needs and disabilities, enabling them to actively participate in social life and utilise their skills and abilities for the betterment of society (Examples 7–13).

**Example 7.** Chen Weijing, associate professor at the Peking Union Medical College, who co-founded ALSOLIFE–a special education platform that helps children with autism master life skills and reduce problematic behaviour–said **inclusive education** gives such children *opportunities* [57].

**Example 8.** "**Inclusive education** has become an important *way to* improve the social adaptability of disabled children and help them participate in public life", Hu said [43].

The media tends to focus on the positive aspects that IE brings to society after implementation (i.e. "opportunities" and "improve the social adaptability" in Examples 7 and 8). The phrase "associate professor at the Peking Union Medical College" is employed as a nomination strategy [14] to portray Chen Weijing and lend credibility to his statement regarding the advantages of IE in Example 7. The adjective "important" is used to depict the significance of IE and serves as a predication strategy [14] in Example 8. Evaluating IE projects provides one of the benchmarks for measuring project effectiveness. This media attention reflects people's concerns regarding the impact of IE projects. The public and stakeholders desire to know whether these projects can genuinely achieve their stated goals and generate positive societal effects. Therefore, evaluating and assessing the success of IE projects becomes an important topic.

**Example 9.** Having access to quality and **inclusive education** *gives* people the knowledge and skills they need to access decent jobs and live more prosperous lives," he stressed [58].

**Example 10.** Hu Meifeng, principal of the special education school in Beichen district, says that more than 80 children from the district have joined the education program in the compulsory education phase, and more than 20 people have found *jobs* after receiving **inclusive education** [43].

IE represents an educational approach with the primary goal of aiding individuals with disabilities, particularly children, in acquiring knowledge and skills, as well as assisting them in securing future employment [18, 59]. Within the framework of IE, schools offer courses and resources that cater to diverse abilities and needs, ensuring that each student is afforded equal learning opportunities. The core principle of IE revolves around individual-centredness, placing emphasis on the distinctive needs and talents of students. It entails the provision of tailored learning plans and support, inclusive of professional guidance and counselling from teachers, to meet the specific requirements of every student. By employing personalised learning approaches, individuals with disabilities can fully realise their potential, adapt, and apply their knowledge and skills, as Example 9 shows. The verbal phrase "gives people the knowledge and skills" functions as a predication strategy [14] to emphasise the advantages of IE that it will provide to students with special needs.

IE not only focuses on the acquisition of knowledge and skills but also aims to promote vocational aptitude among individuals with disabilities. Schools collaborate with communities to provide practical opportunities and vocational training, thereby helping students to secure jobs and realise their personal value, as illustrated by Example 10. The figure "20" out of "more than 80" in Example 10 represents the success rate of people with special needs who participated in the IE program in Beichen district. These statistics serve as an argumentation strategy (specifically a topos of numbers) [14] that the media employed to support its claims and generalisations with statistical evidence, emphasising the growing number of children with special needs who benefit from IE. The media showcases the accomplishments and outcomes of IE by featuring successful cases and personal stories, offering the public a benchmark for assessing the effectiveness of these initiatives.

The learning environment has a significant impact on learning outcomes (Examples 11–13). A positive and supportive learning environment can stimulate students' interest and motivation, leading to improved learning outcomes. Mutual learning promotes the sharing of

knowledge and understanding, where students can complement each other's information and thinking styles through communication and discussions, thereby deepening their understanding of the subject matter.

> **Example 11.** Wang Jincheng, principal of the middle school where Zihe studies, said **inclusive education** *provides* a safer and more inclusive environment for special children, allowing them to be healthier physically and mentally, and to build relations with their peers [43].

> **Example 12.** After a two-month preparation, he started his class, which has three disabled and three able-bodied children, a perfect model of **inclusive education**, where children can *learn* from each other [53].

> **Example 13.** In China, more and more schools and welfare institutions are *promoting* **inclusive education** for children of different ages and with different health conditions, and help those with different functional needs to enjoy equal access to education and integrate into society in a better way [48].

Example 11 reported Wang Jincheng, who believed in the benefits of IE. The description "principal of the middle school" was attributed to him to enhance the credibility of his statement, serving as a predication strategy [14]. Phrases such as "to be healthier physically and mentally" and "build relations with their peers" emphasise the dual advantages of the learning environment for students' psychological and physiological well-being. These phrases serve as a perspectivisation strategy [14], expressing the positive perspectives of IE in assisting children with special needs. According to Example 12, a classroom consisting of "three disabled and three able-bodied children" serves as an ideal model of IE, promoting mutual learning among students. The number "three" is an argumentation strategy (specifically a topos of numbers) to present an inclusive classroom culture that consciously balances the numbers of disabled and able-bodied children [14]. It gives the audience a sense of the Chinese professional community providing equal opportunities for all students and encouraging mutual learning and understanding, particularly benefiting disabled children. The aim of IE is to promote and facilitate the integration of children with disabilities into society, as demonstrated in Example 13. The phrase "enjoy equal access to education and integrate into society in a better way" serves as a perspectivisation strategy [14], showcasing the optimistic views of IE in assisting children with special needs.

In summary, the advantages of IE represent the second discourse. It offers equal educational opportunities and fosters social development and prosperity.

### 4.3. Challenges faced by inclusive education

The final discourse revolves around the challenges faced by IE in China, with only a limited amount (4.8%) of news coverage dedicated to the data (Table 2). Analysis indicates that IE in China is still in its exploratory phase and encounters various challenges [46]. The primary challenges encompass limitations in accessibility and a shortage of well-trained teachers (Examples 14–15).

> **Example 14.** Although encouraged by government, the **inclusive education** for children is almost *nonexistent* in China. Many autistic children still cannot share a mainstream classroom with their peers, even after special training [60].

> **Example 15.** However, according to Li Weihong, vice-president of China Blind People's Association, in special education schools, blind and visually impaired students have little

contact with the school curriculum, while in regular schools where a system of **inclusive education** is used, a *lack* of capable teachers means they don't receive good, specialised and skilled training [61].

The availability of IE resources varies across different regions in China. In more developed areas supported by policies, IE programs are implemented more effectively. However, in certain remote or impoverished areas, schools may encounter a scarcity of resources, resulting in limited access to suitable educational resources for students with disabilities, as Example 14 indicates. In Example 14, the adjective "nonexistent" serves as a predication strategy [14] to describe the level of implementation of IE in China, demonstrating IE's availability or presence. Additionally, news reports draw attention to the issue of inadequate experienced teachers in numerous Chinese schools. For instance, Example 15 shows in particular rural areas, due to factors like uneven distribution of educational resources and mobility of talented individuals, there is a high demand for specially trained and experienced special education teachers. The expression "don't receive good, specialised and skilled training" was employed to depict the teachers, fulfilling the predication function [14]. This presents challenges for schools in providing appropriate support to students with special needs.

## 5. Discussion

From a macro perspective, the English-language news regarding IE in China is relatively limited, and the attention it receives is inadequate. This could be attributed to the fact that China's IE has only received policy support in the last two decades [62]. The findings are consistent with the reality: although China has a historical foundation for IE, influenced by Confucian culture and moral harmony [17], it truly began in the early 1990s. Since 2000, China has actively promoted the nationwide development of IE and implemented various policies and measures. These include the establishment of physical infrastructure in schools to accommodate students with disabilities, teacher training to enhance their skills in IE, and the promotion of public awareness about IE [18].

The first discourse "efforts to develop IE" demonstrates that China's present special education system can be characterised as a three-tier system comprising of the "government-school-individual". The government establishes pertinent policies, schools adhere to regulations, and individuals depend on policy benefits to acquire financial support. In China, numerous schools, such as kindergartens, primary schools, middle schools, and high schools, are publicly administered by the government. Consequently, these schools take proactive measures to accommodate children with special needs in line with the implementation of government policies, which aligns with the standpoint of M. Deng et al. [63]. Our study shows that the media coverage echoes previous research findings, indicating that mainstream teachers in China demonstrate care and compassion and enhance equal opportunities for students' interactions with individuals with disabilities [4, 64]. Our study also reveals that the state-controlled media shows that teachers are committed to ensuring equal educational opportunities for every student. Through positive encouragement and support, educators help students fully realise children's potential [65, 66]. Educators and IE teachers play a key role in driving IE, with the government and schools offering support and guidance, while the true responsibility for implementing IE rests with them.

The Chinese government places great importance on the development of IE due to its advantages, as indicated by the second discourse on the "benefits of IE". Implementing IE can create a more inclusive and friendly learning environment [67]. By dismantling segregation and discrimination, schools can foster interaction and cooperation among students,

promoting social integration. This not only benefits the comprehensive development of students with special needs but also helps other students develop values of respect and understanding towards differences. The media, by reporting successful cases and personal experiences as shown in the results, offers the public an evaluation reference for the effectiveness of IE. Numerous studies and theories have supported the importance of mutual learning in students' development, such as Vygotsky's social development theory [67] and Slavi's [68] research on cooperative learning. In such an environment, students are encouraged and motivated, leading them to actively participate in sports and physical activities, thereby improving their physical health. Additionally, a positive emotional state helps to reduce students' anxiety and stress, promoting their mental well-being. Moreover, when students engage in collaborative learning and cooperation with their peers, they can establish more friendships and support networks, which can help alleviate feelings of loneliness and social difficulties, ultimately enhancing their psychological well-being, as the second discourse indicates.

Promoting and facilitating the integration of children with disabilities into society is the goal of IE [69]. It creates a diverse and inclusive learning environment where children can communicate and collaborate with peers from different backgrounds and abilities. Such experiences help them gain a deeper understanding of and respect for the differences in others, nurturing empathy and an inclusive mindset. Consequently, they become better equipped to assimilate into society. In urban areas and developed regions, certain schools have implemented special education classes or customised learning programs for students with special needs. These measures aid children in their social integration. By offering a diverse and inclusive learning environment that fosters empathy, teamwork, and personal growth, children are more effectively prepared to integrate into society, form positive relationships with others, and lay the groundwork for future achievements. The able-bodied children can observe and grasp the significance of understanding and acceptance from their disabled peers, while the disabled children can seize more learning opportunities through interactions with their able-bodied counterparts. This finding is consistent with Hodkinson's [70] research, which reveals that 85% of the participants expressed a willingness to forge friendships with able-bodied children. The process of mutual learning and exchange aids in cultivating students' empathy, understanding, and amiable attitudes, thereby positively influencing their social and communication skills.

The news media further illustrate the specific challenges that IE faces in China, as indicated by the third discourse on the "challenges faced by IE". According to M. Deng and Poon-McBrayer [17], IE in China began relatively late but has undergone rapid development in the past decade. Nevertheless, China still faces numerous challenges in implementing IE, including a shortage of skilled professionals, inadequate resources, and an imperfect education system [18, 62]. Due to the high demand for special education but limited resources, some children with special needs are unable to access the appropriate support and services they require, which aligns with the findings discussed by M. Deng and Poon-McBrayer [17]. Furthermore, there are concerns regarding the urban-rural gap and regional disparities within China's special education system, as illustrated in the results. S. Liu et al. [54] indicate that children with special needs in remote areas encounter difficulties in accessing the same level of education and support as their urban counterparts [54]. The third discourse highlights the ongoing disparities and obstacles in implementing IE, which necessitate extra resources and experienced educators to offer adequate support and training. In actuality, though, it is essential to undertake thorough research and grasp the local context and policies in various regions and schools in order to formulate more accurate and efficient solutions for IE. Owing to the nature of official news reporting, which predominantly concentrates on positive elements, this aspect is less frequently mentioned.

The three hegemonic discourses employed by the state-controlled media to cultivate a favourable national public image, as uncovered in this research, differ from those examined in prior studies [10, 12, 15]. In contrast to the findings of Yu, Tay, and Yue [12], China was depicted as a victim, a fighter, and a co-operator during the COVID-19 pandemic, using the "victim, fighter, and cooperation discourses". Similarly, Chan and Yu [10] discovered that China was portrayed as a co-operator, a responsible country, and a believer in science through three positive discourses: "cooperation as a win-win solution", "people's lives and well-being as the priority", and "science as the spirit". Likewise, Yu [15] revealed that China was presented as a defender with a "discourse of resistance" against unwarranted foreign hostility regarding the spread of COVID-19. In this study, China was depicted as more of a facilitator, fostering the advancement of IE through the discourses of "efforts to develop IE", "benefits of IE", and "challenges faced by IE ", which pertain specifically to the field of IE. The researchers observed that Chinese English-language news media tend to employ discursive strategies such as nomination, predication, argumentation, and perspectivisation to construct these distinct discourses [14]. Yu [15] also identified the strategies of enemification, victimisation and heroisation in constructing the "discourse of resistance" within the state-controlled media. These various discourses and discursive strategies can be utilised at different times to cultivate a positive national image for China in its state-controlled news media, targeting foreign readers.

## 6. Conclusion

As mentioned at the outset of this article, there have been no studies utilising corpus and discourse analysis to comprehend how IE in China is portrayed internationally to build its national image. This study fills the research gap by employing a corpus-assisted critical discourse analytic approach to investigate how IE in China is portrayed in mainstream official media such as *China Daily* and *Xinhua News Agency*, thus examining the external representation of IE in China. The study identifies three primary discourses: "efforts to develop inclusive education", "benefits of inclusive education", and "challenges faced by inclusive education". The discursive strategies used to construct these discourses include nomination, predication, argumentation, and perspectivisation [14]. The findings reveal a three-tier system of "government-school-individual" in which IE is constructed in China. The study acknowledges the significant efforts made by the government in policymaking, policy implementation, improving learning environments, and investment in IE. Furthermore, we offer a model for conducting corpus-assisted CDA and propose that researchers could investigate recurrent patterns in the media on a significant scale to unveil pertinent socio-cultural phenomena. Due to the specific nature of China's news media, which is controlled by the government, certain issues deserve attention, such as the tendency to highlight achievements while neglecting challenges, resulting in difficulty in addressing existing problems, such as a shortage of well-trained teachers. A limitation of this study lies in that it has only investigated China's English-language news media, which is state-controlled, targeting foreign audiences. Future research could consider investigating non-state-controlled media to compare with our findings.

## Author Contributions

**Conceptualization:** Yating Yu.

**Data curation:** Zhiying Yu.

**Funding acquisition:** Kuen Fung Sin.

**Supervision:** Kuen Fung Sin.

**Writing – original draft:** Gaoqiang Lu.

**Writing – review & editing:** Yating Yu.

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
