## [Decision Letter · Decision Letter 0]

2 Jan 2024

PONE-D-23-29105Media representations of China's inclusive education: A corpus-assisted discourse analysisPLOS ONE

Dear Dr. Yu,

Thank you for submitting your manuscript to PLOS ONE. After careful consideration, we feel that it has merit but does not fully meet PLOS ONE’s publication criteria as it currently stands. Therefore, we invite you to submit a revised version of the manuscript that addresses the points raised during the review process.

We look forward to receiving your revised manuscript.

Kind regards,

Chao Gu

Academic Editor

PLOS ONE

Journal Requirements:

2. Thank you for stating the following in your Competing Interests section: "NO authors have competing interests"

**Additional Editor Comments:**

Please make substantial revisions based on the feedback from the two reviewers.

Reviewers' comments:

Reviewer's Responses to Questions

**Comments to the Author**

1. Is the manuscript technically sound, and do the data support the conclusions?

Reviewer #1: Partly

Reviewer #2: Partly

2. Has the statistical analysis been performed appropriately and rigorously? 

Reviewer #1: N/A

Reviewer #2: Yes

3. Have the authors made all data underlying the findings in their manuscript fully available?

Reviewer #1: Yes

Reviewer #2: Yes

4. Is the manuscript presented in an intelligible fashion and written in standard English?

Reviewer #1: Yes

Reviewer #2: Yes

5. Review Comments to the Author

Reviewer #1: The study addressed an important topic but the manuscript needs to be substantially revised. I suggest that the authors consider addressing the following issues:

1. It is necessary to strengthen the link between the project's objective and the research questions. Why did the analysis focus on linguistic and discourse analysis? As a reader, I would like to see that the research questions are compellingly justified by the writing.

2. I do not think that the literature review connects to the research questions well. Although readers need some contextual understanding, the authors should really elaborate on the argument for the research questions to be addressed. It is alos necessary for the authors to present any conceptual framework that might have underpinned the analysis. Otherwise, it will be a very descriptive analysis.

3. Can the authors be explicit about how each research question was addressed in the analysis of data?

4. I do not see a clear alignment between research questions and findings. It is not clear whether both questions were throughly answered by the results. The results are also very descriptive suggesting that there is a need to go beyond the surface level of analysis (e.g. counting).

5. How do the findings add to the exsiting body of literature on the topic? This is a problem for this reader as the authors did not contextualize the study with the relevant disciplinary field (e.g., is this a study for readers in special education or public communication?). As a result, I am not sure in what sense the findings are signficant new knowledge. The authors should really go back to the introduction and literature review to articulate a clear argument for the study by contextualizing it within a significant disciplinary knowledge gap.

6. Any thoughts on the implications that the study has for readers? Implications for research and practice?

Reviewer #2: This manuscript presents a corpus-assisted discourse analytic research on the Chinese government initiative for inclusive education. Using English-language local news channels, the researchers examined 73 news articles for meticulous line-by-line concordance analysis. The researchers endeavoured to review China’s self-image regarding the initiative of inclusive education. Results have named three major categories (i.e. efforts to develop inclusive education, benefits of inclusive education and challenges before inclusive education). Overall, the paper is adequately executed and the analyses performed are appropriate to answer the research questions. The authors are off to a good start, however, this study requires additional exploration of some news articles that is not belong to “party-controlled news media” platforms for more unbiased findings. Otherwise, this methodological issue can limit the results of the study and the discussion.

Moreover, authors are advised to build their results section on the exhibition of the three main categories they have found. And let the discussion be leaded by their research questions compared to the related literature. Ultimately, the study presenting well-promising idea and adding to the current knowledge. Consequently, I would recommend it for publication with relatively major correction.

The research questions seems to be underdeveloped and need more clarification. For instance, what they mean by ‘the linguistic patterns’ in the second question? However, literature review relatively present an overview of the initiative of inclusive education with little coverage of related literature. Readers need to be convinced by your knowledge gap, so firstly presents what others conducted followed by what you will add. However, you argued that your study will unveil the influence of news media on in shaping foreign public perceptions. I do not think you have explored foreign public perceptions to know the level of influence of these media channels on them.

On methods, your selection of the tool Factiva needs to be justified, what is it? Is it a database rolled by the government or a private sector? Have you tested its validity and reliability? Furthermore, the authors noted that the extract data (i.e. 137 articles) was filtered manually, the possibility of errors being included cannot be ruled out. Therefore, it may be more reliable to use the original electronic data for filtrations. Otherwise, manual filtrations can still be an option by some experts consultation. Another eye for such manual process seems inevitable.

There is a need for segregation between results & discussion sections. Authors should try to represent their results at the findings section and leave discussion and/or comparison to the discussion part. For example, your discussion of examples 11, 12 and 13 on pages 23 & 24 should be moved to the discussion part. In addition, the study fails to address how the findings relate to previous research in this area. The authors should rewrite their discussion to reference the related literature, especially recently published work.

Ultimately, the current manuscript requires significant methodological changes for it to be considered for publication. Overall, the topic explored in this paper is engaging, yet the limitations identified should not be ignored.

Minor corrections

Page 9 at the Introduction- Try to rephrase the sentence starts with ‘However, due to... ’.

Page 11 Paragraph 2- Mention the findings of Florian and Linklater.

Page 13 Paragraph 1- What do you mean by the number 500 after ConcGram?

Page 13 Paragraph 1- At the bottom, mention the page number for the quote.

Page 14 Paragraph 2- You mentioned the category (others) at Table 2, try to provide some examples for that category.

Page 18 Paragraph 1- The sentence “This echoes previous....” And the one followed should be at the discussion part. Please see my prior comment about differentiating between findings’ explanations and discussion.

Page 21 Paragraph 1- Same remark as the previous comment.

Page 25 Paragraph 1- Sentence starts with “This educational model enables...” until the end seems repetitive and through unnecessary, please omit it.

Page 26 Paragraph 2- Move that to the discussion section.

Reference 21 - “2020; 10” Is the 10 referring to article’s pages number? If yes then mention the last page number.

Reference 22 - Same comment the previous one.

Reference 31 - Do you mean China Daily?

Reference 35 - Make sure that is the correct reference style.

Reference 37 - What “100912” mean?

Reference 55 - Unnecessary bracket usage and revise pages number.

6. PLOS authors have the option to publish the peer review history of their article (what does this mean?). If published, this will include your full peer review and any attached files.

Reviewer #1: No

Reviewer #2: No

---

## [Author Response · Author response to Decision Letter 0]

2 Feb 2024

Please see my file "Response to Reviewers".

---

## [Decision Letter · Decision Letter 1]

22 Feb 2024

Media representations of China's inclusive education: A corpus-assisted critical discourse analysis

PONE-D-23-29105R1

Dear Dr. Yu,

We’re pleased to inform you that your manuscript has been judged scientifically suitable for publication and will be formally accepted for publication once it meets all outstanding technical requirements.

Kind regards,

Chao Gu

Academic Editor

PLOS ONE

Reviewers' comments:

Reviewer's Responses to Questions

**Comments to the Author**

1. If the authors have adequately addressed your comments raised in a previous round of review and you feel that this manuscript is now acceptable for publication, you may indicate that here to bypass the “Comments to the Author” section, enter your conflict of interest statement in the “Confidential to Editor” section, and submit your "Accept" recommendation.

Reviewer #1: All comments have been addressed

Reviewer #2: All comments have been addressed

2. Is the manuscript technically sound, and do the data support the conclusions?

Reviewer #1: Yes

Reviewer #2: Yes

3. Has the statistical analysis been performed appropriately and rigorously? 

Reviewer #1: N/A

Reviewer #2: Yes

4. Have the authors made all data underlying the findings in their manuscript fully available?

Reviewer #1: Yes

Reviewer #2: No

5. Is the manuscript presented in an intelligible fashion and written in standard English?

Reviewer #1: Yes

Reviewer #2: Yes

6. Review Comments to the Author

Reviewer #1: The authors have addressed my concerns through revision. I have no further issue with the study. The manuscript can be accepted for publication.

Reviewer #2: (No Response)

7. PLOS authors have the option to publish the peer review history of their article (what does this mean?). If published, this will include your full peer review and any attached files.

Reviewer #1: No

Reviewer #2: No

---

## [Editor Report · Acceptance letter]

19 Mar 2024

PONE-D-23-29105R1 

PLOS ONE

Dear Dr. Yu, 

I'm pleased to inform you that your manuscript has been deemed suitable for publication in PLOS ONE. Congratulations! Your manuscript is now being handed over to our production team.

Kind regards, 

on behalf of

Dr. Chao Gu 

Academic Editor

PLOS ONE